# Rational Usage of Fracture Imaging in Children and Adolescents

**DOI:** 10.3390/diagnostics13030538

**Published:** 2023-02-01

**Authors:** Ralf Kraus, Klaus Dresing

**Affiliations:** 1Department of Trauma and Orthopeadic Surgery, Klinikum Bad Hersfeld, 36251 Bad Hersfeld, Germany; 2Department of Trauma Surgery, Orthopaedics and Plastic Surgery of the University Medical Center Göttingen, 37075 Göttingen, Germany

**Keywords:** pediatric fractures, ALARA, plain radiograph, ultrasound

## Abstract

In this paper, authors introduce the basic prerequisite for rational, targeted, and above all, child-oriented diagnosis of fractures and dislocations in children and adolescents is in-depth prior knowledge of the special features of trauma in the growth age group. This review summarizes the authors’ many years of experience and the state of the current pediatric traumatology literature. It aims to provide recommendations for rational, child-specific diagnostics appropriate to the child, especially for the area of extremity injuries in the growth age. The plain radiograph remains the indispensable standard in diagnosing fractures and dislocations of the musculoskeletal system in childhood and adolescence. Plain radiographs in two planes are the norm, but in certain situations, one plane is sufficient. X-rays of the opposite side in acute diagnostics are obsolete. Images to show consolidation after conservative treatment is rarely necessary. Before metal removal, however, they are indispensable. The upcoming diagnostical tool in pediatric trauma is ultrasound. More and more studies show that in elected injuries and using standardized protocols, fracture ultrasound is as accurate as plain radiographs to detect and control osseous and articular injuries. In acute trauma, CT scans have only a few indications, especially in epiphyseal fractures in adolescents, such as transitional fractures of the distal tibia or coronal shear fractures of the distal humerus. CT protocols must be adapted to children and adolescents to minimize radiation exposure. MRI has no indication in the detection or understanding of acute fractures in infants and children. It has its place in articular injuries of the knee and shoulder to show damage to ligaments, cartilage, and other soft tissues. Furthermore, MRI is useful in cases of remaining pain after trauma without radiological proof of a fracture and in the visualization of premature closure of growth plates after trauma to plan therapy. Several everyday examples of rational diagnostic workflows, as the authors recommend them, are mentioned. The necessity of radiation protection must be taken into consideration.

## 1. Introduction

The basic prerequisite for rational, targeted, and above all, child-oriented diagnosis of fractures and dislocations in children and adolescents is in-depth prior knowledge of the special features of trauma in the growth age group. Since the morphology of injuries in this age group is less dependent on the cause of the injury and the type of impact than in adulthood but rather on the state of maturity of the musculoskeletal system, the pediatric traumatologist is confronted with recurring, stereotypical injury patterns that they must be familiar with.

For the same reason, the history of the course of the injury, although necessary, rarely allows a relevant conclusion to be drawn about the expected injury. It is particularly important when it is necessary to distinguish injuries that are actually accidental from those that are not, for example, when there is a suspicion of child endangerment or even child abuse.

The significance of the physical examination is also limited. Here, it is essential to avoid inflicting additional pain on the injured child through the examination. It is limited to the inspection of the injured body part. It is important, also regarding the subsequent imaging diagnostics, to determine whether or not there is a deformity of the injured body part, especially in the case of an injured extremity. The determination of certain fracture signs, such as crepitations, must be omitted. The examination of a painfully restricted function must also be carried out very cautiously. However, subtle checking of neurovascular function in the periphery is essential, especially if the pediatric traumatologist is aware of the frequency of concomitant injuries to nerves and vessels, such as in elbow injuries.

Through a combination of prior pediatric traumatology knowledge, history reserved physical examination, and expected therapeutic consequences following these, the appropriate imaging modality must then be indicated to confirm or exclude a fracture or dislocation.

The workhorse in the diagnosis of fractures and dislocations in the growth age is still the plain radiograph. A comprehensive study evaluating 28,000 children with emergencies showed that 45.5% of injured children underwent primary radiography after an accident event [1]. X-ray images are widely available in emergency rooms and medical practices, and the result of the examination is available quickly. Diagnosticians and traumatologists are experienced in interpreting the images and thus able to draw therapeutic consequences quickly. Sonography and the cross-sectional imaging procedures CT and MRI are available as complements and alternatives.

However, especially in childhood and adolescence, the effects due to the use of ionizing radiation must be considered [2]. There are many reasons for this. Child tissue has a higher water content than adult tissue, which results in a higher radiation dose required to penetrate a layer of tissue of equal thickness [3]. In addition, it is more radiosensitive due to an increased mitotic rate [4]. Children and adolescents usually have a long life expectancy and, therefore, a long period of time in which malignancies can develop after irradiation [5,6]. In contrast, older adults, in particular, usually die from other causes before irradiation can have an effect. It is further relevant that up to four times more hematopoietic bone marrow is found in the extremities of infants and young children, which are particularly frequently examined with X-rays in connection with injuries [3]. Thus, the benefit-risk ratio should be weighed meticulously, and the indication for X-ray examination should be strict [2]. Therefore, the ALARA principle (as low as reasonably achievable) should be applied [7].

This review summarizes the authors’ many years of experience and the state of the current pediatric traumatology literature. It aims to provide recommendations for rational, child-specific diagnostics appropriate to the child, especially for the area of extremity injuries in the growth age.

This review does not deal with the interpretation of different X-ray, ultrasound, or MRI images but primarily with the indication for the use of diagnostic imaging procedures. Therefore, the authors have intentionally refrained from inserting image material for illustration to not distract from the actual topic.

### 1.1. Plain Radiographs

The plain radiograph remains the indispensable standard in the diagnosis of fractures and dislocations of the musculoskeletal system in childhood and adolescence. The necessities of radiation protection outlined above must be taken into consideration by correct technical implementation with the use of filters, apertures, intensifying screens, etc. The contribution of the pediatric traumatologist to the reduction of radiation exposure consists of the correct indication that is adequate to his question. Some aspects deserve special attention.

### 1.2. One or Two Planes?

Primary diagnosis with two image planes perpendicular to each other is common. Suppose the deformity of the injured region is clinically recognizable, and the first X-ray allows a clear diagnosis from which a definite (surgical) treatment indication can be derived. In that case, it may not be necessary to perform the second plane. This is true for many primary severely displaced fractures, whether they are diaphyseal, metaphyseal, or joint-involving epiphyseal. In the case of highly displaced, i.e., unequivocal fractures to be treated by reduction and osteosynthesis already in one plane, the second plane should be performed in the operating room under anesthesia.

Under no circumstances should the second plane be omitted in the case of clinically absent deformity, but nevertheless, clear secondary fracture signs such as pain, swelling, and functional limitation [8]. It is not uncommon for an undisplaced fracture to be impossible to visualize in only one plane.Likewise, a second plane must not be omitted if a first X-ray image permits an unambiguous diagnosis. Still, the information that is decisive for the therapeutic decision can only be expected from the second plane. This applies, for example, to the extent of antecurvation in supracondylar humerus fractures.Furthermore, a second plane is indispensable if an initial X-ray clearly demonstrates a fracture. Still, the pediatric traumatologist expects a further significant aspect of injury based on his prior knowledge. This is necessary, for example, when an ulnar fracture is demonstrated to exclude concomitant radial head dislocation (Monteggia injury).

### 1.3. X-ray of the Opposite Side

The normal radiographic anatomy of the growing, immature skeleton changes from year to year. The depiction of the meta- and epiphyses with their growth and traction plates, as well as the large bone portions that are only cartilaginous and thus cannot be depicted radiographically, require a high degree of prior knowledge in the interpretation of the image material. Without this prior knowledge, the question may arise whether the findings are normal for the age of the patient or the result of an injury. In this situation, it is not uncommon even today to resort to the supposedly easiest way out and take plain radiographs of the uninjured opposite side of the affected child in order to obtain a comparison with the current normal condition. It is the firm conviction of the authors that this approach should be strictly rejected and considered obsolete [9].

To try to find out about the normal anatomy of the injured child first does not correspond to the above-mentioned requirement for the pediatric traumatologist to enter into the diagnosis and therapy of the same with deep anatomical prior knowledge. In cases of doubt, atlases or electronic aids can be consulted.The use of ionizing radiation on the part of the body that is not injured or diseased does not fulfill the mandatory existence of a “justifying indication”.Finally, X-raying the opposite side offers no guarantee that the injury at hand will actually be detected. And it offers no guarantee that the appropriate therapeutic conclusions will then be drawn for the injured extremity.

Therefore, if doubts remain about the presence of an injury or its extent after the X-ray images of the injured body region, the switch to another diagnostic method should rather be sought. This may be an ultrasound examination or, in rare cases, a cross-sectional imaging procedure.

Exception: There are indeed rare indications for x-raying the opposite side, for example, if a typical, congenital, bony deformity is discovered as an incidental finding, which is known to occur not infrequently bilaterally. An example of this is congenital, proximal, and radioulnar synostosis. However, even in this case, x-rays of the opposite side may not serve to “understand” the lesion.

### 1.4. Radiographs after Therapy

If active therapy in the form of reduction and, if necessary, osteosynthesis has been performed in the case of a displaced fracture or dislocation, the result must be documented by imaging in every case. Forensic and medicolegal aspects should play less of a role. Rather, the medical question should be answered as to whether the therapeutic goal that was formulated before the treatment and was to be achieved with the manipulation that took place was actually achieved.

In the case of a large number of stable compression or torus fractures in children, on the other hand, X-ray control after the application of immobilizing bandages can, and indeed must, be dispensed with.

### 1.5. Consolidation Control

If no manipulation in the form of reduction or osteosynthesis has been performed on an injured bone in children and adolescents, it may initially be assumed that bony healing is taking place safely and without a doubt. The few exceptions to this rule, in which the natural course may lead to pseudarthrosis (e.g., ulnar epicondyle, radial humeral condyle), must be known to the pediatric traumatologist. In all other cases, the determination of bony consolidation can be made by clinical examination alone. This applies, for example, to the common fractures of the distal forearm and clavicle. Here, after a reasonable period of time, it is the callus free of pressure pain and the return of pain-free function that reliably indicates bony healing.

The situation is different after active reduction and osteosynthesis, whether performed percutaneously or openly. Manipulation or insertion of foreign material can disrupt, at least delay, or even prevent otherwise safe fracture healing. Therefore, under these circumstances, radiographic consolidation control cannot yet be dispensed with.

### 1.6. X-ray and Metal Removal

Before a material removal, the changing indication also in childhood and adolescence is not to be discussed here. The sufficient bony consolidation and, thus, the indispensable prerequisite for the intervention must be proven by imaging for the aforementioned reasons. Here, the necessary X-ray diagnostics must not be dispensed with. In addition to proving the actual healing of the fracture, it is also important to detect possible complications in advance, which may be associated with an extension of the surgical intervention (e.g., screw or wire fracture, wire migration, bony overgrowth).

Every experienced pediatric traumatologist knows that, especially in these cases, imaging using image intensifiers is sometimes indispensable, even during surgical metal removal.

In contrast, there is no regular indication for radiographs for documentation after metal removal, according to the authors. Bony consolidation had to be proven before the procedure, and the surgeon must be able to judge the completeness of the metal removal on the basis of the removed materials themselves. On the other hand, especially in the case of unsuspected retention of osteosynthesis material, postoperative documentation is useful.

### 1.7. Ultrasound

Further developed from arthrosonography, fracture sonography has found its way into the spectrum of pediatric traumatology diagnostics. Numerous papers demonstrate the useful application of sonography in acute traumatology in children and adolescents [10,11,12,13,14]. Fracture diagnosis may have advantages over radiography in selected injuries in terms of lack of radiation exposure, repeatability, and simultaneous soft tissue assessment [15].

In fractures, disruption or distortion of the bone surface, which is rich in reflection, can be seen. The periosteum can usually be identified by the double-layered image structure. In fracture, it may be raised by hematoma, and the pattern may be spread. The growth plate, which has not yet closed, can be visualized. The bone axis can be safely assessed in the appropriate body regions [16]. In distal forearm fractures in children 0 to 12 years of age, studied with the Wrist Safe algorithm in 6 standard projections, showed that safe diagnostic and therapeutic management is possible with sonography [17].

Ultrasonography is being used more and more in emergency diagnostics as point-of-care ultrasonography (POCUS), not only in the abdominal and thoracic regions but also in the extremities [18,19]. Here, it serves as a screening method to initiate additional diagnostics in a targeted and focused manner [20].

In many places, however, the view still prevails that ultrasound diagnosis of fractures is unreliable and dependent on the examiner [21]. Neither the reliability of the findings has been sufficiently scientifically reproduced nor the safe use of fracture ultrasonography has been widely ensured. Other authors counter that the best radiation protection in childhood and adolescence is physicians who are optimally trained in sonography and use this technique [3]. There can be no doubt about the increasing importance of fracture sonography in growing-age traumatology in the future [7].

The current literature offers a variety of observations and studies on the use of sonography in pediatric traumatology.

### 1.8. Skull

Sonography is an integral part of pediatric emergency medical practice in suspected skull fractures and in young children with intracranial injuries. It has superseded the skull x-ray [22,23].

### 1.9. Upper Extremity

Clavicle: Ultrasound diagnosis is sometimes considered more useful than radiography in clavicle fractures. This is especially true in children younger than 10 years of age, in whom therapy is generally conservative and consists solely of pain management [14,24,25,26,27].

Proximal humerus: Sonography may also prove superior to conventional radiography in proximal humerus fractures. A standard protocol in 4 planes should be followed. Under this, axial deviation may be more accurately measured than on radiographs [28,29].

Distal humerus/elbow: Essential for proving injury is the visualization of a joint effusion. If this is missing, further radiographic examinations can be omitted in up to 48% of cases [30]. The diagnosis of a supracondylar humerus fracture can be made on ultrasound from a cortical prominence or a cortical gap. The sensitivity is reported to be 100%, and the specificity is 93.5% [10]. Sonography allows early differentiation between hanging and complete fractures in primary undisplaced radial condyle fractures [31]. Diagnosis of radial neck fractures with ultrasound was one of the first applications [32]. The sensitivity is high [33]. Therapy management using sonography has since been described [34].

Distal forearm: Fracture detection at the distal forearm is successful with sonography with a sensitivity of 94% to 96% and a specificity of 92% to 97% and can thus be used validly for fracture diagnosis [35,36,37,38]. The examination is recommended in 6 standard sectional planes (radius longitudinally from palmar, radial, and dorsal, ulna longitudinally from dorsal, ulnar, and palmar) [39].

### 1.10. Lower Extremity

Hip: Sonography has long been established in pediatric orthopedics for the detection and classification of hip dysplasia. In addition, it is used to detect hip effusion in conditions such as coxitis fugax and septic coxitis, as well as contour irregularities in Perthes disease or epiphysiolysis capitis femoris [40,41].

Femur: In femoral fractures in infancy treated conservatively with traction, it has been shown that follow-up ultrasound examinations can significantly reduce the number of radiographic examinations [42].

Ankle: After radiographic exclusion of bony injuries, sonography has also proven useful for detecting ligamentous injuries [43,44].

### 1.11. Occult Fractures

In the absence of evidence of fractures on radiographs but with a typical clinic, ultrasound is suitable for detecting injuries, including those to the small bones of the hand [45,46]. Even in the case of the nonspecific clinic and clinically unclear fracture localization, ultrasonography was able to detect fractures very validly with sensitivity (93.2%) and specificity (99.5%) in a study of 653 children [47].

### 1.12. CT

Computed tomography is not a substitute for knowledge of the recurrent, stereotypic injury patterns typical of childhood and occurring well into adolescence. A prerequisite for the use of computed tomography in this age group is the existence and use of special, radiation-saving examination protocols.

Computed tomography may be useful in individual cases of acute fractures in the joint region in adolescents. This applies, for example, to transitional fractures of the distal tibia if they cannot be adequately traced on conventional radiography. In the upper extremity, the rare coronal shearing injuries of the capitulum and trochlea humeri regularly require computed tomography for treatment planning.

Even fractures of the carpus and tarsus in adolescents often require cross-sectional CT imaging for adequate treatment planning [48].

The indications for shock room CT are more stringent in severely injured children and adolescents than in adults. While in many cases, targeted diagnosis based on the individual injury pattern is sufficient, whole-body CT must be used for acutely life-threatening injuries and injury combinations [49,50].

Another important indication for CT in the elective field is the planning of corrective osteotomies in cases of remaining deformities or post-traumatic growth disorders.

### 1.13. MRI

The authors do not see indications for MRI in extremity injuries of growing age in the initial diagnosis to exclude or visualize fractures, but rather subacute to exclude internal joint injuries (e.g., cruciate ligament, meniscus, cartilage) or secondarily in persistent complaints in the region of injury without radiological evidence of a fracture (bone bruise, algodystrophy). Late secondary MRI has its place in pediatric traumatology for imaging bony bridges across the growth plate in clinically diagnosed, post-traumatic, inhibitory growth disturbance, among others. MRI, on the other hand, is not necessary for the diagnosis of a fresh fracture; like CT, it does not replace the specific prior knowledge of pediatric traumatology of the diagnostician and practitioner.

However, MRI is urgently indicated in the acute phase after a spinal trauma with neurological deficits, especially when X-ray and CT cannot show a bony injury as the cause (SCIWORA syndrome). It is then important to exclude, for example, intraspinal hemorrhage [51,52].

Especially regarding MRI, it should be noted that it is not uncommon for a multitude of irrelevant, non-target findings to result, which confuse and unsettle [53].

### 1.14. Application Examples

In the following, some case examples, without claiming to be exhaustive, will further explain the above principles of diagnostics in clinical application.

4-year-old girl, fall on trampoline, pain left shoulder with limitation of movement. Clinical findings: mild swelling and palpable deformity clavicle. Imaging diagnosis: ultrasound, at most X-ray in 1 plane: low angulated clavicle shaft fracture. Therapy: immobilization for pain management, no imaging controls.6-year-old boy, fall from climbing frame. Severe pain right arm. Clinical findings: severe swelling and marked deformity of the elbow. No neurovascular deficit. Imaging diagnosis: x-ray in 1 plane: completely displaced supracondylar humerus fracture type IV according to von Laer. Therapy: rapid reduction and osteosynthesis, radiological documentation of the treatment result and radiological control before metal removal.14-year-old boy, bicycle fall. Pain and immobility of right elbow. Clinical findings: moderate soft tissue swelling at the elbow, pressure pain over both epicondyles. Imaging diagnosis: X-ray of elbow in 2 planes: suspected coronary shear fracture of the capitulum humeri. Supplementary CT for exact fracture presentation and therapy planning. Therapy: open reduction and osteosynthesis, radiological documentation of the treatment result and radiological control before metal removal.9-year-old girl, fall on the playground. Pain in left wrist. Clinical findings: hardly any swelling, moderate pain on movement. Imaging diagnosis: ultrasound: distal metaphyseal radial torus fracture. If necessary, X-ray images in 2 planes to verify the US findings. Therapy: immobilization for pain management. No radiological controls, if parents wish US after end of immobilization if necessary.15-year-old boy, knee torsion during soccer game. Pain in left knee joint and inability to bear weight. Clinical findings: minor joint effusion, pressure pain tibial tuberosity, range of motion and ligament stability not testable due to pain. Imaging diagnosis: X-ray in 2 planes: Exclusion of fracture. Therapy: immobilization, relief, clinical control. In the early course, MRI to exclude an internal knee injury, especially cruciate ligament rupture.2-year-old girl, spontaneous limping on the right side without accident. Clinical findings: pain localization most likely lower leg shaft, no swelling, no deformity. Imaging diagnosis: X-ray lower leg with knee and ankle in 2 planes: no evidence of fracture, no osteolysis. Ultrasound: cortical disruption of tibial shaft with detachment of periosteum. Therapy: under the diagnosis of toddlers fracture, rest, if necessary short-term immobilization. Clinical controls, no X-ray control, ultrasound if parents wish.10-year-old girl, fall while skateboarding. Pain in left ankle. Clinical findings: significant soft tissue swelling and moderate deformity. No neurovascular deficit. Imaging diagnosis: x-ray in 2 planes: Salter-Harris II growth plate displacement with antecurvation of 20 degrees. Therapy: rapid reduction and osteosynthesis, radiological documentation of the treatment result and radiological control before metal removal. Long-term clinical controls. In case of secondary varus deformity: MRI to visualize a suspected bony bridge over the growth plate and therapy planning.

## 2. Conclusions

To successfully diagnose a fracture in the growth age, which is appropriate for the target and the child, the treating pediatric traumatologist must first have extensive prior knowledge. The growth-specific, physiological changes of the skeleton, as well as the stereotypical injury patterns of the still-immature bone and their age-specific changes, must be known.

Clinical diagnosis is limited to narrowing down the injured body region, identifying or ruling out deformities, and examining the periphery of the injury for neurovascular deficits.

From knowledge of the age, affected body region, and clinical findings, a working diagnosis is then made regarding the injury at hand. This is followed by an analysis of which available imaging technique is best suited to confirm or exclude the suspected injury quickly and according to the situation to be able to initiate the first therapeutic steps. With very few exceptions, this will be the X-ray or ultrasound. Secondarily, other procedures can be used if necessary. A specific question must always be formulated, and it must be clarified whether the diagnostic procedure is also capable of answering the question.

The pediatric traumatologist also needs deeply founded knowledge about the further course of growth-specific injuries to be able to offer meaningful clinical and imaging controls. This includes, for example, knowledge of which types of injuries are likely to have an increased incidence of relevant post-traumatic growth disorders.

## Data Availability

Not applicable.

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
