# Peer review of "Rational Usage of Fracture Imaging in Children and Adolescents"

_diagnostics, 2023, doi:10.3390/diagnostics13030538_

Round 1

Reviewer 1 Report

Some of the comments are integrated in the PDF

Author Response

Content and concept of the paper as the changes after review were consented with the guest editor, Dr. Sinan Bakir, Berlin, Germany.

  1. Abstract should be re written.

Some changes have been made.

  1. In last paragraph of the introduction authors should explain a brief passage about theoriginality of their work.
    A review is rarely original, is it? As mentioned the paper bases on personal experience and recent relevant literature.
  2. Define the advantages and disadvantages of X-Rays in children.
    The advantages and disadvantages are explicitly dealt with in lines 47-67 and seem detailed enough to the authors.
  3. What about the consequences of occult fracture in children and old man compare it briefly.

Children after injury with complaints are taken seriously. If no primary evidence of fracture is obtained radiologically, further clarification can be obtained with Point of Care Ultrasound (POCUS). Before an MRI examination, which usually requires anesthesia, is ordered, the child should be reexamined clinically in a timely manner if there is minimal trauma, and then the decision to re-image should be made (e.g., lines 207-208, 278-282).

  1. 5. Conclusion and future work is not well written.

The conclusion section summarizes, what the authors consider to be important and correct. This paper is not about future work, but, it presents the current state of knowledge (see above).

Reviewer 2 Report

Title: Rational usage of fracture imaging in children and adolescents

In this paper author introduce the basic prerequisite for rational, targeted, and above all child-oriented diagnosis of fractures and dislocations in children and adolescents is in-depth prior knowledge of the special features of trauma in the growth age group. This review summarizes the authors' many years of experience and the state of the current pediatric traumatologic literature and aims to provide recommendations for rational, child-specific diagnostics appropriate to the child, especially for the area of extremity injuries in the growth age. The plain radiograph remains the indispensable standard in the diagnosis of fractures and dislocations of the musculoskeletal system in childhood and adolescence. The necessities of radiation protection must be taken into consideration.

This paper is written very well have a good scientific contribution all the results are written very well.

I have the following suggestions. After these changes I will strongly recommend it for publication.

1. Abstract should be re written.

2. In last paragraph of the introduction authors should explain a brief passage about the
originality of their work.

3. Define the advantages and disadvantages of X-Rays in children.

4. What about the consequences of occult fracture in children and old man compare it briefly.

5. Conclusion and future work is not well written.

After these changes I will recommend it for publication.

Author Response

Content and concept of the paper is consented with the guest editor.

  1. Abstract should be re written.

Some changes have been made.

  1. In last paragraph of the introduction authors should explain a brief passage about theoriginality of their work.
    A review is rarely original, is it? As mentioned the paper bases on personal experience and recent relevant literature.
  2. Define the advantages and disadvantages of X-Rays in children.
    The advantages and disadvantages are explicitly dealt with in lines 47-67 and seem detailed enough to the authors.
  3. What about the consequences of occult fracture in children and old man compare it briefly.

Children after injury with complaints are taken seriously. If no primary evidence of fracture is obtained radiologically, further clarification can be obtained with Point of Care Ultrasound (POCUS). Before an MRI examination, which usually requires anesthesia, is ordered, the child should be reexamined clinically in a timely manner if there is minimal trauma, and then the decision to re-image should be made (e.g., lines 207-208, 278-282).

  1. 5. Conclusion and future work is not well written.

The conclusion section summarizes, what the authors consider to be important and correct. This paper is not about future work, but, it presents the current state of knowledge (see above).